# Extracellular Vesicles as Signaling Mediators and Disease Biomarkers across Biological Barriers

**DOI:** 10.3390/ijms21072514

**Published:** 2020-04-04

**Authors:** Pasquale Simeone, Giuseppina Bologna, Paola Lanuti, Laura Pierdomenico, Maria Teresa Guagnano, Damiana Pieragostino, Piero Del Boccio, Daniele Vergara, Marco Marchisio, Sebastiano Miscia, Renato Mariani-Costantini

**Affiliations:** 1Department of Medicine and Aging Sciences, University “G. d’Annunzio”, Chieti-Pescara, 66100 Chieti, Italy; simeone.pasquale@gmail.com (P.S.); giuseppina.bologna@hotmail.it (G.B.); laura.pierdomenico@unich.it (L.P.); mariateresa.guagnano@unich.it (M.T.G.); m.marchisio@unich.it (M.M.); s.miscia@unich.it (S.M.); 2Center for Advanced Studies and Technology (C.A.S.T.), University “G. d’Annunzio”, Chieti-Pescara, 66100 Chieti, Italy; damiana.pieragostino@unich.it (D.P.); piero.delboccio@unich.it (P.D.B.); renato.mariani@unich.it (R.M.-C.); 3Department of Medical, Oral and Biotechnological Sciences, University “G. d’Annunzio”, Chieti-Pescara, 66100 Chieti, Italy; 4Department of Pharmacy, University ‘‘G. d’Annunzio’’, Chieti-Pescara, 66100 Chieti, Italy; 5Department of Biological and Environmental Sciences and Technologies, University of Salento, 73100 Lecce, Italy; daniele.vergara@unisalento.it; 6Laboratory of Clinical Proteomics, “Giovanni Paolo II” Hospital, 73100 ASL-Lecce, Italy

**Keywords:** extracellular vesicles, biomarkers, liquid biopsy, biological barriers

## Abstract

Extracellular vesicles act as shuttle vectors or signal transducers that can deliver specific biological information and have progressively emerged as key regulators of organized communities of cells within multicellular organisms in health and disease. Here, we survey the evolutionary origin, general characteristics, and biological significance of extracellular vesicles as mediators of intercellular signaling, discuss the various subtypes of extracellular vesicles thus far described and the principal methodological approaches to their study, and review the role of extracellular vesicles in tumorigenesis, immunity, non-synaptic neural communication, vascular-neural communication through the blood-brain barrier, renal pathophysiology, and embryo-fetal/maternal communication through the placenta.

## 1. Introduction

### 1.1. Evolutionary Background 

Communication is a central aspect of life; therefore, its evolutionary history dates to the earliest forms of living organisms on Earth. Primordial single-celled organisms likely developed intercellular communication via chemical signals already during the Archean Eon, over 3.5 billion years ago, an ability that likely allowed the formation of microbial mats, attested in the fossil record by accretionary structures (“stromatolites”) formed by cyanobacterial layers [1]. Later, during the Proterozoic Eon (about 2.5 billion to 541 million years ago), the progressive increase in oxygen levels allowed the evolution and spread of the eukaryotes, capable of cellular respiration, a key adaptation based on mitochondria-dependent catabolic reactions that yield abundant biochemical energy [2]. Increased energy availability allowed ever more complex cellular functions and closer interactions among cells [3], culminating in the emergence of multicellular organisms, a watershed event in the history of life that took place during the pre-Ediacaran period of the upper Proterozoic (_~_ 1300–600 million years ago) [4]. The ensuing wide radiation of large multicellular forms of life, well evident in the Ediacaran (about 635–541 million years ago), clearly required the establishment of intercellular communication mechanisms granting the delivery of integrated and undiluted biological messages at discrete distances, even to cells residing in specialized tissues or organs protected by physical barriers (e.g., tight junctions, basement membranes, etc.) [5]. This ability was at the basis of evolution of the three kingdoms of multicellular life, including plants and fungi, which do not possess canonical nervous systems, and animals, where quick communication integrated at the organismal level was further ensured by the development of nervous systems relying on chemical and electrical synapses [4,6]. All in all, this long evolutionary background in deep time accounts for the complexities and redundancies of the communication networks in multicellular organisms. 

The release of biological signals within membrane-bound extracellular vesicles (EVs) may have originated from the need of eliminating aging or damaged plasma membrane (PM) components [7] and may have also been connected with the development of the eukaryotic responses to cell damage. In fact, when a buildup of intracellular debris overwhelms proteasomal and autophagic clearance, the shedding of damaged structural components contributes to cell survival [8,9]. Subsequently, this functional response module proved serviceable in sending warning signals to distant cells, so that EVs became critically important as mediators of intercellular communication, particularly in the Metazoans, where the release of biological messages within stable extracellular vesicles (EVs) delimited by the plasma membrane (PM) lipid bilayer is facilitated by the lack of rigid cell walls.

### 1.2. General Characteristics and Biological Significance of EVs 

EVs are produced constitutively or upon activation, due to inflammation, hypoxia, oxidative stress, shear stress, senescence, cell death, exposure to bacterial endo/exotoxins, uremia, etc. [10,11,12,13,14,15]. The cargo varies depending on type and differentiation of the parent cell, microenvironmental variables, and agents that triggers EV release. Cargo contents include lipid mediators (e.g., eicosanoids), proteins (e.g., cytokines, chemokines, growth factors or other mediators of signal transduction), genetic material (e.g., mRNAs, long/short noncoding RNAs, nuclear and mt DNA) and, in the case of larger vesicles, whole organelles (e.g., mitochondria) [16,17,18]. While the mechanisms that mediate the biological effects of EVs on their cellular targets remain poorly known, it is clear that EVs are implicated in most, if not all, physiopathological processes, including signal transduction, cell growth, and differentiation, metabolic regulation, embryofetal development, organogenesis, tissue homeostasis and repair/regeneration, antigen presentation and immune response, ageing, pathogen-host interactions, carcinogenesis, tumor invasion/metastasis, cardiovascular dysfunction, etc. [9,19,20,21,22,23,24,25,26,27,28,29,30,31,32,33,34,35,36,37]. The EV cargo, packaged within relatively stable membrane-bound structures, is sheltered from degradation by the extracellular enzymes present in biological fluids, and may therefore maintain biological stability over comparatively long periods of time [38]. To assess stability, Sokolova et al. analyzed EVs isolated from various cell types (human embryonic kidney 293 cells, HEK293T; endothelial-colony forming cells and mesenchymal stem cells), and found that EV size and integrity strongly depended on storage conditions: EV diameters significantly diminished within 2 days at 37 °C and 4 days at 4 °C, while storage at −20 °C did not affect size [39]. Kalra et al. investigated the stability of EVs isolated from LIM 1863 colorectal cancer cells [40]. EVs spiked in plasma and stored at 37 °C, 4 °C, −20 °C, and −80 °C were analyzed by Western blot (WB) at various time points for the EV marker TSG101 [41]. Samples stored at +4 °C, −20 °C, and −80 °C resulted TSG101-positive up to 90 days, indicating stability for at least 3 months, with better preservation at −80 °C, while plasma kept at 37 °C conserved TSG101 up to 30 days. With regard to functionality, EVs labeled with the PKH67 green fluorescent dye were normally internalized by colorectal cancer cells LIM 1215 after storage in spiked plasma at −20 °C for 30 days [40].

Due to their substantial stability, EVs circulate systemically and have been detected in basically all body fluids, including blood, urine, cerebrospinal fluid, saliva, milk, and tears [16,42,43,44,45,46,47,48]. Moreover, there is clear indication that EVs can cross multiple biological barriers, as demonstrated by the finding of glial/neuronal EVs in the cerebrospinal fluid, blood, tears, and urine [44]. Specific delivery to distant targets is ensured by cell/tissue tropisms depending on surface markers [49,50]. 

It has been hypothesized that the differences in terms of EV dimensions and surface molecules affect their ability to recognize and capture their recipient cell. In this context, micropinocytosis is theoretically compatible with the target interaction of exosomes and small EVs, but not of large EVs. For larger EVs, it has been demonstrated that EVs themselves may produce their effects by interacting with extracellular specific molecules of their recipient cells or by fusion of the EV membrane with the target cell plasma membrane [51,52]. EVs can also be internalized, through multiple routes that involve endocytosis by both clathrin-dependent and clathrin-independent pathways, producing the final transfer of the EV content into their target cells [53,54]. However, it must be underlined that these mechanisms of EV uptake and cargo delivery are still incompletely characterized [55]. In any case, the first step of EV uptake involves targeting the acceptor cell. Some papers demonstrated that this phenomenon is highly specific and possibly depends both on the phenotype of the EV subtype and the extracellular characteristics of the acceptor cells [56,57]. In this context, the exposure of the integrin CD47 protects EVs from phagocytosis, increasing the EV circulating time in the blood [58]. It is unclear whether further EV uptake into the related acceptor cell occurs through non-specific processes, or through specific, receptor-dependent pathways [59]. Some other works demonstrated, instead, that the EV-cell interaction is totally unspecific and stochastic [60,61]. Additionally, it has been observed that the EV-mediated cargo transfer can occur in an interspecies context [62]. Hence, the EV cargo, which may comprise integrated sets of biological information, is transmitted undiluted and undamaged, and the effects produced can be very strong, even at remarkable distances from the cell of origin [17,18]. Message delivery may be limited to direct activation of PM receptors of the recipient cells or may require internalization via endocytosis or fusion.

### 1.3. EV Subtypes 

Three EV subtypes, namely exosomes, microvesicles (MVs), and apoptotic bodies, are traditionally identified based on size and biogenesis [63,64]. 

### 1.4. Exosomes 

Exosomes, with diameters ranging from 30 to 150 nm, represent the smallest EV compartment, in the size range of viruses [64,65,66,67,68], which, in the form of mature enveloped virions, are in essence exosomes whose cargo and release are determined by viral genetic instructions [69]. Physiologically, exosomal vesicles originate within the lumens of acidic endocytic organelles termed multivesicular bodies (MVBs), which can be regulated by either “endosomal sorting complex required for transport” (ESCRT) proteins or ESCRT-independent mechanisms involving membrane lipids and tetraspanins, phylogenetically-conserved membrane-spanning proteins controlling PM dynamics and vesicle biogenesis [65,70]. MVB vesicles can either be degraded (when MVBs fuse with lysosomes) or released by exocytosis (when MVBs fuse with the PM). Additionally, it is clear that the inhibition of exosome secretion will increase MVB lysosomal degradation and *vice versa* [71]. 

Several pathways have been described to regulate both lysosome function and exosome secretion [71]. Among them, it has been demonstrated that ceramide, through the local production of its metabolite sphingosine-1-phosphate (S1P) [72], is essential for exosome secretion. It is possible that S1P plays a pivotal role in the mechanisms related to the sorting of intraluminal vesicles (ILVa) into MVBs (therefore inducing the exosome release) or into lysosomes that induce their degradation [72]. It is not clear if this sorting is realized at the single ILV level or involves the entire MVB. It is also not known whether the signaling regulating ILV biogenesis determined their fate [71]. Some ESCRT proteins (ALIX, HRS, and TSG101) have been pointed out for their role in autophagy and exosome secretion [73,74]. The direct involvement of the autophagy-lysosome pathway in the exosome secretion has been also demonstrated [75] and even the metabolic status can be involved in the sorting of MVBs [76].

Tetraspanin enrichment, clearly demonstrated by the immunoelectron microscopic analysis of exosomes [77,78], may be necessary for exosome release, as suggested by evidence from knockouts for tetraspanin CD9 [79]. Tetraspanins were also found to be involved in exosomal sorting of cargo molecules, such as the MHC-I/MHC-II immune recognition complexes [80,81] and mRNAs/miRNAs [82,83].

As established in a position paper of the International Society of Extracellular Vesicles (ISEV), exosomes are univocally identified by tetraspanins CD9, CD63, and CD81 [84]. Interestingly, these tetraspanins play critical roles in development, tumorigenesis, and tissue regeneration/repair. CD9, together with another tetraspanin, CD82, is implicated in the exosomal export of β-catenin, and thus modulates Wnt signaling, an ancient developmental pathway critically involved in cell fate determination and organogenesis/tumorigenesis [79]. Furthermore, CD9 controls membrane recruitment of metalloproteinases, such CD10 and ADAM17, and may thus promote cell migration and invasion [85,86]. CD81 is implicated in tumor-stroma interactions [87], and CD63 in melanogenesis [70], which plays a key role in the protection from UV-induced cell damage [88]. 

Recently, Zhang et al. revised exosome subclassification using asymmetric flow field-flow fractionation (AF4), which highlighted subpopulations of large and small exosomes (Exo-L, 90–120 nm; Exo-S, 60–80 nm) and a distinct subpopulation of non-membranous nanosized exosomes, designated ‘exomeres’ (~35 nm), which are the most abundant particles secreted by cancer cells. These three subsets of exosomes have quite specific biochemical and biophysical profiles and definite biodistribution patterns, suggesting distinct functional specializations [89]. This indicates that much remains to be understood about the morphological and functional heterogeneity of the exosomal vesicles. 

As noted above, the hijacking of exosomal pathways contributes to virus spread, as it provides handy exit and entry machinery and affords protection from extracellular enzymes, complement-mediated lysis, and immune responses to capsid antigens [90]. Thus, a better understanding of the relationships between viruses and exosomes might lead to the identification of novel targets for antiviral therapies [91], while the techniques now under development for the study of exosomes will likely result in breakthroughs in diagnostic virology.

### 1.5. Microvesicles and Apoptotic Bodies 

Microvesicles (MVs) generally range from 100 to 1000 nm in diameter and are released by budding or blebbing from lipid rafts or caveolar PM microdomains. MVs released by cancer cells, also designated “oncosomes,” include a larger subtype of vesicles, termed large oncosomes (1–10 µm in diameter). These vehiculate enzymes involved in glucose, glutamine and amino acid metabolism, mitochondrial constituents, mitochondria-derived vesicles [92], and genomic/mitochondrial DNA from the tumor of origin [93]. Oncosomes may therefore modulate the metabolic and genetic potential of their target cells; additionally, they may confer proteolytic activity, promoting invasion/migration, and may influence organotropic metastatic spread [34,35,94], a process that may involve integrin signaling [34].

Importantly, a subset of MVs exposes phosphatidylserine on the membrane surface [64,95]. This occurs in EVs originating from cells activated by stressors, where calcium influx switches on enzymes, such as floppase and scramblase, that flop phosphatidylserines to the outer leaflet of the PM bilayer. Surface phosphatidylserine is a signal for recognition and uptake by adjacent cells, particularly professional phagocytes [96]; therefore, the half-life of these MVs is generally short. The distinctive biogenetic process accounts for the fact that MVs can be readily sub-classified based on annexin V positivity, restricted to MVs that expose phosphatidylserine, and patterns of surface markers, which generally reflect those of the parental cells [27,33,47]. However, phosphatidylserine is also exposed on apoptotic bodies, which are larger vesicles specifically formed during the late stages of apoptosis [97,98]. Apoptotic bodies can be distinguished from other phosphatidylserine-positive MVs based on positivity for caspases 3 and 7 and their substrates (e.g., ROCK1 and PANX1) [99], while size is unreliable, because apoptotic bodies (~0.1 to ~5 µm) overlap in size with other EV subtypes [99]. Here, it should be remarked that phosphatidylserine exposure, although often used as a surrogate marker of apoptosis [100], does not distinguish apoptotic bodies from other MVs [84,99]. 

Lastly, it should be mentioned that several cell types, and resting platelets, secrete respiratory competent mitochondria susceptible of autonomous extracellular signaling as well as of intercellular transfer, a finding of which the implications are still unclear but that could widely expand the scope of cell-cell communication biology [101]. To conclude, classifications mainly based on size do not fit the heterogeneity of the EV populations and their overlaps in cargo, biodistribution, and functions [102]. Thus, in the position paper, the ISEV has endorsed the use of the term “extracellular vesicle” for all EV types, with a generic subclassification as small, if within 100 nm, and medium/large, if above 100–200 nm [84]. We will henceforward include under the “EV” umbrella acronym both exosomes and MVs, excluding apoptotic bodies. 

### 1.6. EVs as Diagnostic and Prognostic Biomarkers

During the last decade, EVs have been pointed out as reliable biomarkers for the diagnosis and the monitoring of human diseases [103]. Being found in different body fluids (blood, urine, bile, bronchoalveolar fluid, saliva), EVs dynamically reflect the status of the disease, by carrying a specific cargo, which stem from the related parental cell and that consist of proteins, miRNAs, mRNAs, long noncoding RNAs (lncRNAs), and lipids. Therefore, EVs are excellent candidates as a source of biomarkers [104,105,106]. It has been demonstrated that EV cargoes, reflecting the pathophysiological characteristics of the cell of origin, result in much more specific than the biofluid where they circulate, given that the latter contains contain less-relevant molecules [107]. At the same time, the EV content displaying dynamic disease-specific information has a high potential even in the development of new prognostic biomarkers. EVs have been already proposed as reliable diagnostic/prognostic biomarkers in many clinical settings, such as coronary artery disease [108], renal [109], liver [110], neurodegenerative [107], and autoimmune [111] diseases, as well as in systemic sclerosis [112], in urological [113], hepatobiliary [103], and hematological [114] malignances, and in breast [115], lung [116], and ovarian cancers [117], and in cancer care [118].

### 1.7. EVs as Drug Carriers

The possible role of EVs as an attractive source of drug delivery vehicles has been largely emphasized [119,120]. EVs present several advantages with respect to some similar drug delivery systems (i.e., polymeric nanoparticles and liposomes) [121], given that they produce limited systemic side effects, while being characterized by a great specificity [122]. The EVs’ potential as drug carriers is inherent in their ability to deliver different types of cargoes, such as interfering RNA (siRNA) or pharmaceutically active molecules, even at long distances [123,124]. It has been demonstrated, in fact, that EVs have high stability in the blood, allowing them to travel long distances within the body under both physiological and pathological conditions. Furthermore, EVs, because of their small size and their animal origins, can avoid phagocytosis, to release their content into target cells, and bypass the lysosome engulfment. Given that EVs are a natural product of the body, they produce a low immune response [121]. On the other hand, their hydrophilic core makes them suitable carriers of water-soluble drugs [125]. The use of EVs to deliver different therapeutics has been already tested, both *in vitro* and *in vivo*, for the treatment of different pathologies, such as cancer and immunological and neurological diseases (reviewed and cited in [126]). In these studies, EVs have demonstrated a high delivery potential and minimal levels of immunogenicity and toxicity. Recent studies were focused on the improvement of both the delivery and the biodistribution of EV content. In these cases, EVs were engineered by anchoring, on EV surfaces, some peptides able to recognize specific receptors on the target cells [127,128,129]. One of the first technologies to generate targeted EVs is called “surface display” and requires genetic modification of the secreting cells [119,130].

## 2. Methodological Approaches

Several techniques can be used for EV identification and characterization [84,102,131]. Specific EV proteins are commonly detectable by immunoblotting for EV-associated proteins (e.g., ALIX, TSG101 and specific tetraspanins, such as CD63, CD9, and CD81), frequently used to confirm the presence of EVs in fluids [84,132]. The most direct method to determine EV size and morphology is electron microscopy (EM), including both transmission EM (TEM) [133,134] and scanning EM (SEM) [39,135,136]. To avoid sample dehydration, cryogenic EM techniques have been developed, of which cryogenic TEM (cryo-TEM) is the most appropriate [137] (Figure 1). Atomic Force Microscopy (AFM), which is also used [138,139], provides information about specific properties, such as EV stiffness and elasticity [140]. While EM and AFM resolve the size of individual EVs, dynamic light scattering (DLS) is used to determine the collective mobility (diffusion coefficient) of the scattering vesicles. The resulting size distribution is characterized by average size and polydispersity [141]. Other techniques include nanoparticle tracking Analysis (NTA), tunable resistive pulse sensing (TRPS), and asymmetric-flow field-flow fractionation (AF4). NTA is based on recording a time-lapse of particles undergoing Brownian motion measuring scattered light (Sc-NTA) or emitted fluorescence (Fl-NTA) [142]. By the analysis of many individual EV trajectories, NTA can assess particle concentration and size distribution, even in polydisperse samples. TRPS detects individual EVs by measuring changes in electrical current as each vesicle passes through an adjustable nanopore [143,144]. AF4 separates EVs on the basis of hydrodynamic size and can identify and sort out vesicles ranging from few nanometers to an undefined level of micrometers [145]. Last but not least, flow cytometry (FC), commonly used for the analysis of cells, is being actively developed for EV analysis [146,147] and is adopted by an increasing number of research groups, mainly to study the larger EVs [148,149]. Direct FC analysis using tracers that stain the whole vital EV compartment, such as lipophilic carbocyanine dyes, combined with phalloidin, that selectively binds to F-actin, accurately discriminates EVs from artifacts [42,43,44,150]. Standardized FC has a high potential for the detection of EVs in body fluids and, when combined with specific antibodies, concurrently allows EV immunophenotypic characterization [42,44,84,131,151,152] (Figure 2). Imaging flow cytometers combining conventional FC with fluorescence provide new, highly sensitive tools for EV studies [153,154].

### 2.1. EVs in Tumorigenesis

The EVs released by cancer cells have been intensely studied and play key roles in the organization of the tumor microenvironment. For example, they are implicated in the reprogramming of normal stromal fibroblasts to activated cancer-associated fibroblasts (CAFs) [155,156,157,158,159]. In turn, CAFs secrete EVs that boost tumorigenesis by instigating metabolic changes, proliferation, epithelial-mesenchymal transition (EMT), motility, and migration in cancer cells, endothelial cells, and other stromal fibroblasts [160,161,162,163]. Tumor-derived EVs also vehiculate pro-angiogenic molecules that promote endothelial proliferation, migration, cell–cell adhesion, and vascular tube formation [164,165]. All this may reflect the deregulation of physiological functions exerted by EV-mediated intercellular communication during organogenesis and tissue regeneration/repair [166,167,168]. In fact, EV-mediated interactions between inflammatory cells, endothelial cells, mesenchymal cells and tissue-specific stem cells are involved in the regeneration of injured tissues and organs, including skeletal muscle, cardiac muscle, gastrointestinal, neural, renal, and respiratory tissues [169,170,171,172,173].

### 2.2. EVs and Immunity

EVs are also critical to the regulation of innate and adaptive immunity [174]. For example, regarding innate immunity, amplification of the chemotactic recruitment of neutrophils during the inflammatory response is mediated by an autocrine/paracrine cascade driven by leukotriene B4 vehiculated within EVs released by chemotactic neutrophils [175]. Additionally, it has been demonstrated recently that the secretion of EV-embedded mitochondrial components and even whole mitochondria by activated monocytes induces systemic proinflammatory type I IFN and TNF responses in endothelia [176]. This response may contribute to cardiovascular disease and to other autoinflammatory diseases associated with type I IFN and TNF signaling. Adding a layer of complexity, EVs from mesenchymal stem cells can inhibit the macrophage-mediated immune response through the transfer of miRNAs, such as *miR-451a*, *miR-1202*, *miR-630*, and *miR-638*, that target Toll-like receptor signaling and the NF-kB pathway [177]. Therefore, MSC-EVs appear to be a promising cell therapy in various human diseases, including the possibility to play a pivotal role as immunotherapeutic agents [178]. With regard to adaptive immunity, it is well known that MHC class II-restricted antigen presentation depends on vesicular transport [179]. Furthermore, EVs deriving from dendritic cells promote antigen-specific T-cell activation [180,181] and T helper (Th) differentiation towards the Th1 phenotype [182]. Lastly, it has been shown that T regulatory cells quench Th1 (CD4+ IFNgamma+) inflammatory responses by EV-mediated transfer of miRNAs suppressing Th1 cell proliferation and cytokine secretion [183].

### 2.3. EVs and the Nervous System

Intercellular communication based on vesicle exo/endocytosis culminates at chemical synapses, a necessity for metazoans, which need to coordinate complex behaviors and physiological functions via quick signals at the organismal level. Nonetheless, the evolutionary origin of chemical synapses is rooted in the widespread and less specialized forms of vesicle-mediated interactions considered in the present review, which paved the way for this development [184]. In this regard, it is interesting that EVs released from nerve terminals promote synaptic growth at developing larval neuromuscular junctions of *Drosophila melanogaster,* although they do not directly affect synaptic signaling [54,185,186]. EVs have been also implicated in glial-neuronal communication. Studies on rats indicate that EVs from Schwann cells promote the regeneration of peripheral nerve axons [171], both *in vitro* and *in vivo*, possibly through EV-mediated inhibition of RHOA, a suppressor of axon regeneration, and transfer of Schwann cells-derived ribosomes [187]. In the central nervous system, EVs secreted by oligodendrocytes have been associated with enhanced neuronal viability and increased neuron firing rates [188,189]. Less it is known about the role of EVs in neuron-neuron communication, although recently, neuronal EVs carrying Ephrin B2 have been shown to induce the collapse of the growth cone, a critical component of axon pathfinding [190]. Moreover, a wide piece of literature has been dedicated to the role of EVs in neurodegeneration, since EVs are involved in many diseases, such as Alzheimer’s, progressive multiple sclerosis, amyotrophic lateral sclerosis, and Huntington’s disease [191].

### 2.4. EVs and the Blood-Brain Barrier

The blood–brain barrier (BBB) is a term used to describe the unique properties of the microvasculature of the central nervous system (CNS). CNS vessels are continuous non-fenestrated vessels, presenting specific properties that allow the tight regulation of the movement of molecules, ions, EVs, and even cells between the blood and the CNS. In essence, the BBB is composed of a monolayer of specialized endothelial cells characterized by high-resistance tight junctions (TJs) that limit the flux of solutes. Slight differences between blood-spinal cord barrier and BBB have been reported [192], but for our purpose we will not treat them separately. A blood–cerebrospinal fluid barrier, at the level of the choroid plexuses, and a blood–leptomeningeal barrier, at the level of the subarachnoid microvessels, have also been described [193,194]. Furthermore, the epithelioid pia mater acts as an additional CNS barrier that regulates solute and cell traffic between the CSF in the subarachnoid space (SAS) and the sub-pial parenchyma [195,196]. Outside the CNS, a blood–retinal barrier has been also described. It consists of retinal pigment epithelial cells and retinal capillary endothelial cells, tightly connected by specialized junctional complexes, similar to those found at the BBB level [197]. In the CNS, the BBB and the neighboring cells (pericytes, perivascular astrocytes, microglia, and neurons) form a functional unit called neurovascular unit (NVU) [198]. EVs serve as major mediators of the inter-cellular crosstalk within the NVU [57,194]. This physiologically contributes to neural differentiation and synapsis formation, but deregulation of such communication may be highly detrimental to nervous system function [199,200]. The EVs deriving from CNS endothelia have been largely studied. They may act in both paracrine and autocrine manners. The paracrine action can be direct when EV surface molecules interact with specific membrane receptors on target cells, or indirect, when mediated by interactions between EV cargo proteins and cell pathways, after EV internalization [201]. It has been demonstrated that EVs originating from different endothelial contexts (aortic, brain, umbilical) protect oligodendrocytes from apoptosis and may promote myelination processes [202]. These could be generalized functions of the EVs of endothelial derivation. In addition, several *in vivo* studies pointed out the effects of EVs from brain microvascular endothelial cells (BMECs) on neighboring NVU cells, such as pericytes and astrocytes [194]. The uptake of BMEC-derived EVs by pericytes was shown by Yamamoto et al., who demonstrated that stimulation with inflammatory cytokines and endotoxin induces immediate, dose-dependent shedding of EVs from BMECs. These EVs, when added to the culture medium of cerebrovascular pericytes, are rapidly internalized and cause upregulation of VEGF-B mRNA and protein, an effect likely mediated by miRNAs vehiculated in the EV cargo [203]. In contrast, a few studies have shown positive effects using BMEC EVs produced under basal or non-inflammatory conditions [202,204]. It is evident that EVs can cross the BBB in both directions, propagating inflammation across the blood brain barrier (BBB), mediating neuroprotection, and modulating regenerative processes [45] (Figure 3). In fact, EVs shed from CNS neurons and glia can be found in peripheral blood and in tears [44,205], particularly when inflammatory mediators activate CNS endothelia, enhancing BBB permeability [206,207]. In this regard, it has been observed that the permeability of the BBB is increased in neurodegenerative conditions such as Alzheimer’s disease, vascular dementia, and multiple sclerosis [45,208,209,210]. Furthermore, systemic inflammation increases BMEC permeability and modulates BBB functions [207,211,212]. Ajikumar et al. demonstrated that activated neutrophils release EVs that are internalized by the human cerebral microvascular endothelial cell line hCMEC/D3 via energy-dependent mechanisms (endocytosis and micropinocytosis). EV uptake significantly altered the transcriptomic profile of hCMEC/D3 cells, dysregulating genes associated with tight junctions, ubiquitin-mediated proteolysis, and vesicular transport and resulted in a significant increase of cell membrane permeability and in a decrease of trans-endothelial electrical resistance [213]. At the NVU level, EVs can mediate the communication between immune cells and the cerebral vasculature [214]. Monocytes activated by IFNα and/or lipopolysaccharide generate small EVs carrying microRNAs (e.g., *miR-222*, *miR-155*, *miR-146a*, *miR-146b*, and *miR-125a-5p*), that boost monocyte chemotaxis by triggering an NF-kB-mediated inflammatory response in microvascular endothelia. Thus, the inhibition of EV release by monocytes reduces CNS inflammation [214].

Studies conducted on metastatic brain cancer models strongly suggest that EVs produced by cancer cells affect the BBB, paving the way for a preferential metastatic diffusion to the CNS [66]. EVs from the 831-BrT subline of the triple negative MDA-MB-231 breast cancer cell line [66], which metastasizes to the CNS, interact mainly with CD31-positive brain endothelial cells [66] via the β3 integrin subunit (CD61), associated with the EMT, migration, and metastasis. It has also been shown that cancer-derived EVs enhance BBB permeability via the transfer of *miR-105*, which targets the tight junction protein ZO-1 in BMECs, inducing loss of cell–cell adhesion. Tominaga et al. [215] further demonstrated that cancer-derived EVs disrupt the BBB through the delivery of *miR-181c*, which down-regulates *PDPK1* gene with a consequent decrease in phosphorylated cofilin and disorganization of the actin cytoskeleton, followed by the destabilization of tight junctions in BMECs [215].

As in the case of trans-renal release, EVs could pass through the BBB *in vivo* by transcytosis mechanism [216,217,218], normally minimal in CNS endothelia [218,219] (Figure 3). In fact, the inhibition of clathrin-dependent endocytosis with chlorpromazine [220] and/or ML141, a CDC42/RAC1 GTPase inhibitor [221], significantly decreased EVs uptake by BMECs [218]. Inhibition of macropinocytosis by 5-(*N*-ethyl-*N*-isopropyl) amiloride (EIPA) and cytochalasin D [220] had similar effects [218]. Following endocytosis, CNS-tropic EVs colocalized with EEA1, a marker of early endosomes [222] and with RAB11 [218], a marker of recycling endosomes [223,224]. Further studies demonstrated colocalizations with SNARE complexes implicated in endosomal recycling, exocytosis, late endosome-lysosome fusion, and vesicular fusion with the basolateral membrane [218,225]. Overall, it can be concluded that while a subpopulation of CNS-tropic EVs is sorted into late endosomes for degradation, a large subset is sorted into RAB11+ recycling endosomes for release at the basolateral membrane [218]. These processes are physiologically controlled by regulatory mechanisms disrupted under pathological conditions [217,218]. MSC-derived exosomes are able to cross the BBB, and many evidence demonstrated that MSC-derived exosomes can re-induce self-tolerance, lowering the subsequent complications with respect to other treatments. Therefore, therapeutic applications of MSC-derived exosomes are contributing to core advances in the field of autoimmune diseases [226].

### 2.5. EVs in Tears

Tears are a complex mixture of molecules, water, and salts. A recent proteomics study has identified 1526 proteins in tears [227], revealing that tears are less complex as a body fluid than serum or plasma. On the other hand, tears represent an easy and accessible biological fluid. Therefore, the study of tear composition has been proposed as an ideal source for discovering biomarkers. In this context, the presence of small EVs has been demonstrated in tears [228]. Extracellular Vesicles in tears may carry relevant information related to the status of eyes and the other bordering structures. It has been recently demonstrated that EVs in tears of primary open-angle glaucoma patients (POAG) carry a specific pro-inflammatory protein cargo, possibly participating to POAG pathophysiology [43]. Interestingly, a number of evidences suggested that tear composition may reflect the health of the CNS [229]. The study of tears was first suggested for multiple sclerosis (MS) diagnosis [230,231]. Since then, different investigators have performed tear studies in order to assess new biomarkers of the disease. Since the presence of IgG oligoclonal bands in cerebrospinal fluid (CSF) represent a marker of MS, in order to obtain a less invasive biomarker, the presence of IgG oligoclonal bands was also searched in tears of patients [231,232,233]. Aside from oligoclonal bands, it was recently demonstrated the presence in tears of neural-derived and microglial-derived EVs and it was shown that the molecular cargo of EVs circulating in tears of MS patients highly overlapped with that of CSF EVs, indicating an EV-mediated molecular link between CSF and tears and suggesting the ability of EVs to deliver information from the CNS into tears [44]. The mechanism explaining this phenomenon was never described, but it should be remembered that the eye and nervous system tissues have the same embryological origin.

### 2.6. EVs in Renal Pathophysiology

EVs act as intercellular messengers within the nephron and collecting duct and are constitutively shed into the glomerular filtrate by several types of kidney cells, particularly podocytes and proximal/distal tubular and collecting duct epithelia [234,235]. Electron microscopy shows that podocyte-derived EVs adhere to the brush border of target proximal tubular epithelial cells and are readily internalized in concentration-dependent manner [235,236]. Cellular stress, which occurs in a variety of systemic or local pathological conditions that affect the kidneys, produces profound changes in EV features [237,238,239,240]. For example, the release of EVs in the glomerular filtrate is actively induced by mechanical stretch and hyperglycemia, and this provides an early urinary marker of glomerular injury in hypertension and diabetic nephropathy. At the molecular level, the mechanisms that activate stress-induced EV biogenesis in the kidney seem to be multifactorial, as EV release cannot be directly induced by specific stress effectors, such as angiotensin II or TGF-β, alone [241]. Podocyte-derived EVs fuse with proximal tubular epithelial cells, inducing phosphorylation of P38 and SMAD3, expression of extracellular matrix proteins, such as fibronectin and collagen type IV, and consequent tubular fibrosis, which contributes to loss of renal functions [242].

Notably, EVs released in the glomerular filtrate can be retrieved from the urine, together with EVs shed by the epithelia of the urogenital tract, and thus, have a high potential as non-invasive biomarkers for clinical conditions involving the urinary system [38,243,244,245]. Degradation of urinary EVs begins within two hours of urine collection, and for optimal EV preservation, urine should be stored at −80 °C with protease inhibitors [246]. Transcriptomic analysis can help to determine the origin of the urinary EVs. In fact, specific mRNAs, such as those coding for podocin (glomerulus), cubilin (proximal tubule), and aquaporin 2 (collecting ducts) [234], indicate the origin from definite regions of the nephron and collecting ducts. Comprehensive proteomic studies contribute to the identification of the origins of urinary EVs [247,248].

The kidneys are characterized by high blood flow, required to maintain adequate delivery of plasma for glomerular filtration, a key process in organismal homeostasis. The blood that flows through the kidneys contains abundant EVs, mostly originating from endothelial cells, platelets, erythrocytes, and leukocytes, particularly in the presence of systemic diseases. Under physiological conditions, these EVs would not be expected to pass the highly efficient glomerular filtration barrier, which is formed by three layers: 1) fenestrated endothelial cells; 2) glomerular basement membrane (GBM); 3) podocytes [243,249,250]. In fact, while the fenestrated endothelium is permeable to macromolecules and nano-sized particles, the GBM consists of a three-dimensional meshwork of fine fibrils forming evenly sized pores with diameters of 2.5–2.8 nm [251], much smaller than the diameter of the smallest EVs (about 30 nm). Furthermore, the podocytes, which reside in the visceral layer of Bowman’s capsule and form the final filtration barrier, are connected to the GBM by interdigitated foot processes that leave open only thin gaps, designated filtration slits, for filtration through the GBM. These slits, however, are coated by a diaphragm of filter proteins, such as podocin, nephrins, podocalyxin, and protocadherins, which further reduce the filtration surface [250,252]. Thus, glomerular filtration is highly selective: only molecular complexes below 6.4 nm in diameter and under 70 kDa may physiologically transit into the lumen of the nephron [253].

Despite the considerations stated above, experimental evidence, still mechanistically unexplained, shows that EVs introduced in the systemic circulation can reach the urine [254]. In a rat model, fluorescently labeled EVs injected into the aorta were identifiable in kidney sections, suggesting a bypass of the glomerular filtration barrier, as proven by the retrieval of labeled EVs from urine at 12 h from the injection. These urinary EVs seemed to be functional, as indicated by ready uptake from HEK293 cells. Trans-renal EV release could depend on a process similar to transcytosis, i.e., vesicular uptake followed by transcellular release, a well-known pathway for the selective trans-endothelial transport of plasma albumin and low-density lipoproteins [216,217].

In renal cell carcinoma (RCC), a different lipid composition in urinary exosomes has been identified through lipidomics investigations [255].

Trans-renal EV passage is facilitated under pathological conditions. In diabetic nephropathy, for example, the meshwork structure of the glomerular basement membrane is loosened, which results in the formation of pores and tunnels with diameters up to 10–80 nm or even larger [256]. These pores allow the passage of EVs released from glomerular endothelial cells, which thus reach the podocytes. These endothelial EVs may vehiculate *TGF-β1* mRNA, a key mediator of the epithelial-mesenchymal transition, whose translation activates myofibroblastic intracapsular proliferation and glomerular dysfunction. Thus, an EV-mediated endothelial-podocyte crosstalk contributes to glomerular fibrosis and loss of renal function in diabetic nephropathy [252].

Urinary EVs are obviously attractive for their potential translational relevance to the diagnosis and monitoring of renal or genitourinary diseases, and, possibly, even of systemic diseases. However, further development is needed to move towards routine clinical use. In particular, additional research is necessary to improve our understanding of trans-renal EV release mechanisms in health and disease.

### 2.7. Placental EVs

It has been demonstrated recently that the trophoblast and syncytiotrophoblast, major components of the chorionic membrane and well-known mediators of embryo-feto-maternal communication, are a relevant source of placenta-derived EVs [257]. These EVs are recognizable based on the exposure of placental alkaline phosphatase (PLAP) [258,259] and are connected with the occurrence of trophoblast deportation, i.e., the shedding of trophoblast/syncytiototrophoblast clumps into maternal blood [260]. Following release, they circulate in placental blood, enter the uterine veins, and can be found in the systemic maternal circulation [257,259]. In addition, trophoblastic EVs can be retrieved from the amniotic fluid [261,262].

Interestingly, *miR-210*, one of the most expressed placental miRNAs, strongly linked with the hypoxia pathway, is vehiculated within placental EVs [263]. This miRNA participates in the EMT, vasculo/angiogenesis and cell migration, and could contribute to maternal endothelial dysfunction, a condition linked to eclampsia, a life-threatening emergency in pregnancy. This seems to be also the case for PLAP-positive trophoblastic EVs expressing EGFR [264].

Several papers discuss the role of EVs in feto-maternal crosstalk during pregnancy. In particular, EVs play relevant roles in modulating maternal immunity during pregnancy. In this scenario, immunomodulation aims at maintaining efficiency of pathogen elimination without harming embryofetal development [257,265]. The secretion of EVs may contribute to explain how the placenta evades the cytotoxic effect of the maternal immune system, modulating, at the same time, the immune tolerance to fetal antigens. Fas ligand (FASL) and TNF-related apoptosis-inducing ligand (TRAIL) are vehiculated by syncytiotrophoblastic EVs and mediate T cell apoptosis, enhancing immunotolerance. In addition, placental EVs increase T-dependent IFNγ production [266]. Maternal monocytes bind and internalize placental EVs [266,267,268] that induce the release of cytokines related to “type 2” immunity, such as TNFα, MIP-1α, IL-1α, IL-1β, IL-6, IL-8, while cytokines involved in “type 1” immunity are down-modulated [267]. Thus, placental EVs promote a ‘type 2′ skewed immunity [267]. Furthermore, it has recently been shown that placental MSC-Derived EVs can promote myelin regeneration in animal models of multiple sclerosis, probably by controlling the immune response as during pregnancy [269].

Three complement regulatory proteins, membrane cofactor protein (MCP; CD46), decay-accelerating factor (DAF; CD55), and protectin (CD59), are represented in the cargo of trophoblastic EVs. DAF is a phosphatidylinositol (PI)-anchored protein, localized to the brush border of the syncytiotrophoblast [270]. Both DAF and MCP regulate T cell functions independently from complement regulatory roles. These proteins may prevent complement activation, while EVs are free-floating in the maternal circulation and may regulate the function of maternal T cells, which might otherwise engage into immune responses against paternally derived placental antigens [271].

The pattern of placental EV secretion is markedly modified in pathological conditions, such as preeclampsia and gestational diabetes mellitus (GDM). Placental EV release increases in preeclampsia, and the EVs induce inflammatory, anti-angiogenic, and pro-coagulant effects [272]. The increased release of placental EVs, together with the rise of free fetal hemoglobin, may cause endothelial cell re-programming, systemic increase of the arterial smooth muscle tone, and hypertension [272,273]. Interestingly, total-miRNAs and *miR-210* vehiculated within EVs were found increased in preeclamptic compared to healthy pregnancies [274]. Higher concentrations of placental EVs, with lower PLAP+/total EVs ratios, were also demonstrated in GDM patients compared to healthy pregnant women [275]. The EV compartment of GDM patients vehiculates a number of miRNAs (e.g., *miR‒122-5p*; *miR‒132-3p*; *miR‒1323*; *miR‒136-5p*; *miR‒182-3p*; *miR‒210-3p*; *miR‒29a-3p*; *miR‒29b-3p*; *miR‒342-3p*, and *miR-520h*) involved in cell proliferation, trophoblast differentiation, insulin secretion and regulation, and glucose transport [276]. Overall, it now seems evident that placental EVs are of critical relevance for the control of the maternal immune response and can contribute to the induction of severe metabolic and vascular complications of pregnancy.

## 3. Conclusions

In conclusion, EVs represent an ideal source of diagnostic and prognostic biomarkers for liquid biopsy [277,278]. Given their ability to pass through biological barriers, EVs, which are easily obtainable from accessible biological fluids, such as tears, blood, and/or urine, can provide valuable information about pathophysiological conditions that affect organs or systems that are inaccessible or not easily accessible to direct biological sampling, such as CNS, kidneys, and embryo-fetal placental tissues. Furthermore, the inhibition of extracellular vesicles formation and release might lead to novel therapeutic approaches valuable for the control of CNS inflammation, metastatic tumor spread, renal insufficiency eclampsia, and so on. On the other hand, EVs might be usable to vehiculate targeted biological therapies [279]. Further developments in this research field of critical translational relevance are expected and eagerly awaited.

## Figures and Tables

**Figure 1 ijms-21-02514-f001:**
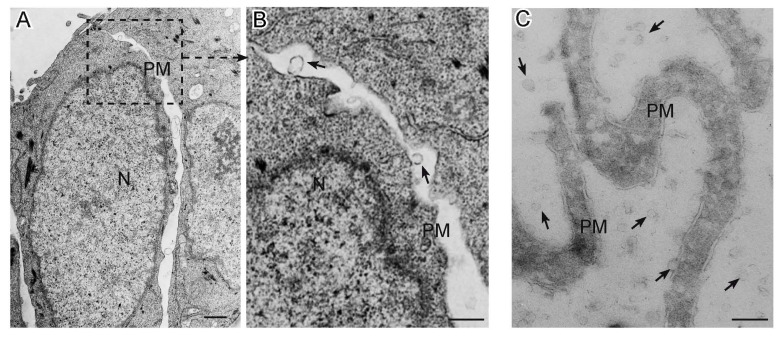
Extracellular Vesicles released by cancer cells as seen by electron microscopy. Panel (**A**) shows a transmission electron micrograph of a section of SW480 colorectal cancer cells, with panel (**B**) detailing microvesicles (arrows) budding from the plasma membrane. Panel (**C**) presents a cryo-electron microscopic section of microvilli from the SKBR3 breast cancer cell line, which release by budding abundant extracellular vesicles (arrows) in the culture medium. N: nucleus; PM: plasma membrane; bars: A, 1 µm; B–C, 0.5 µm.

**Figure 2 ijms-21-02514-f002:**
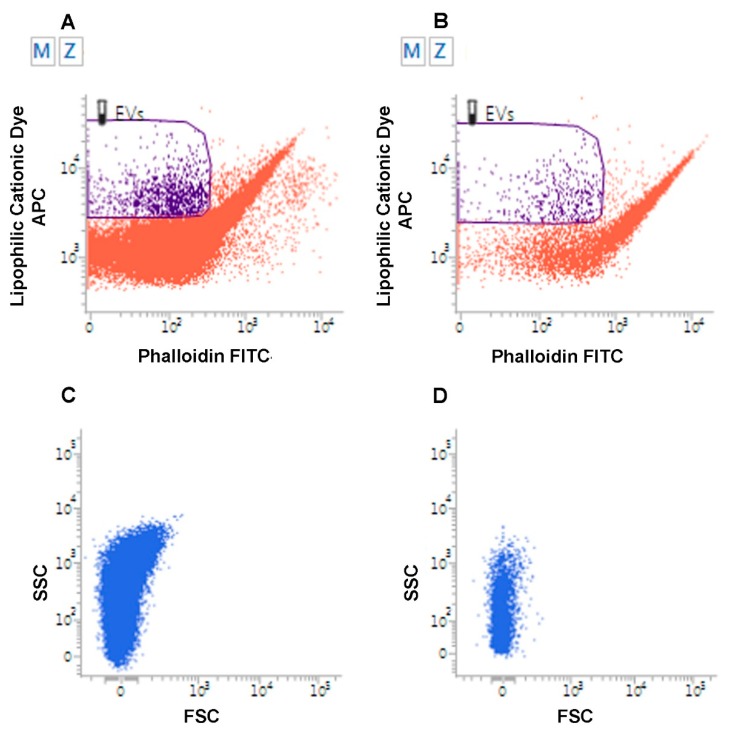
Flow Cytometry analysis of Extracellular Vesicles. Flow cytometry analysis of EVs from urine samples from a patient affected by low grade superficial papillary carcinoma of the urinary bladder (**A**) and from a healthy volunteer (**B**). EVs are identified as lipophilic cationic dye-positive (LCD) particles emitting on the allophycocyanin-APC-channel, and negative for phalloidin, emitting on the fluorescein isothiocyanate (FITC) channel. Dot density in the EV gate is higher in the urine of the patient. (**C–D**). Flow cytometry analysis of EVs from urine of patient affected with metastatic pheochromocytoma (**C**) relative to a healthy volunteer (**D**). EVs were analyzed for their scattered parameters on Forward Scatter (FSC)/Side Scatter (SSC) dot-plots. Number of events in the EV gate is higher in the urine of the patient.

**Figure 3 ijms-21-02514-f003:**
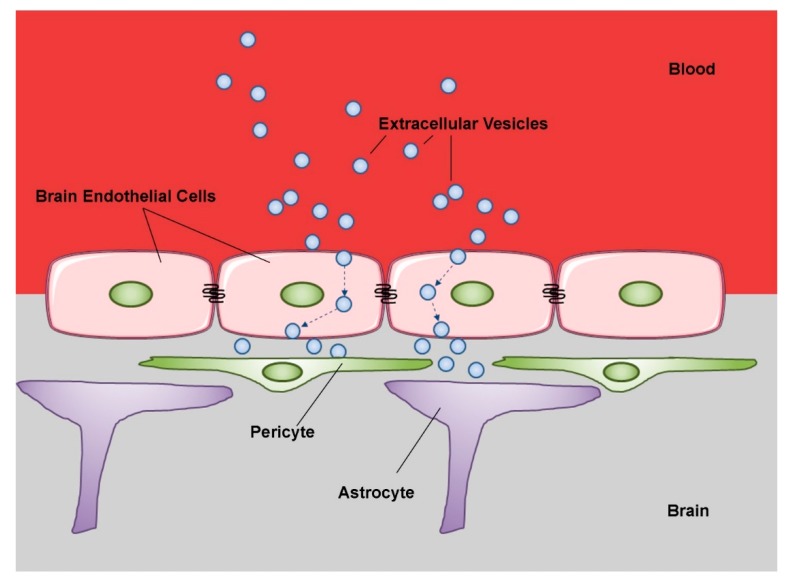
Schematic representation of extracellular vesicle transcytosis through the blood brain barrier (BBB). The BBB is a complex physical barrier made up by endothelial cells in close association with pericytes and astrocytes. Extracellular vesicles circulating in peripheral blood first cross through transcytosis the tightly joined endothelial cells of the cerebral capillaries. Then, the vesicles interact with the next cell layers, pericytes and astrocytes. These sequential transfers imply mechanisms of selective recognition, trans-cellular transport and release. (modified from Servier Medical Art, licensed under a Creative Common Attribution 3.0 Unported License; http://smart.servier.com).

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
