# Peer review of "Extracellular Vesicles as Signaling Mediators and Disease Biomarkers across Biological Barriers"

_ijms, 2020, doi:10.3390/ijms21072514_

Round 1

Reviewer 1 Report

The review by Simeone et al"Extracellular vesicles as signaling mediators and disease biomarkers across biological barriers" described the biological significance of extracellular vesicles as mediators of intercellular signaling, discuss the various subtypes of extracellular vesicles and the principal methodological approaches to their study, and review the role of extracellular vesicles in tumorigenesis, immunity, non-synaptic neural communication, vascular-neural communication through the blood-brain barrier, renal pathophysiology and embryo-fetal/maternal communication through the placenta.

The review is very well written and covered various aspects of EVs across biological barriers. Since eye is also another biological barrier, it would be better to cover the role of EVs in eyes as well.

Author Response

Reviewer 1

The review by Simeone et al"Extracellular vesicles as signaling mediators and disease biomarkers across biological barriers" described the biological significance of extracellular vesicles as mediators of intercellular signaling, discuss the various subtypes of extracellular vesicles and the principal methodological approaches to their study, and review the role of extracellular vesicles in tumorigenesis, immunity, non-synaptic neural communication, vascular-neural communication through the blood-brain barrier, renal pathophysiology and embryo-fetal/maternal communication through the placenta.

RQ1: The review is very well written and covered various aspects of EVs across biological barriers. Since eye is also another biological barrier, it would be better to cover the role of EVs in eyes as well.

A1: Thanking the Reviewer for the suggestion, we dedicated a new section to the presence of EVs in tears.

Reviewer 2 Report

In the current review article,  authors shave summarized the role of extracellular vesicles in intercellular communications, its diagnostic and prognostic roles. The manuscript is written well, however, authors need to address the following pieces of information. 

1. Authors have described the size of exosomes ranges from 30 nm to 100 nm while several reports suggest the range 30-150 nm ( Hoshino et al, 2015 Nat.; Costa-Silva et al 2015 NCB, Patel et al., 2017, British J Cancer, etc).

2. In the EVs subtype, authors did not describe Apoptotic Bodies and specific markers for the ABs and Microvesicles, if known just like exosomal markers.

3. Authors also need to describe how MVBs are getting escaped from the lysosomal degradation.

4. How EVs mediate cellular communication? What are the different modes of interaction and internalization of exosomes?

5. How these EVs can be utilized as diagnostic, prognostic and drug carriers?

Author Response

Reviewer 2

In the current review article, authors shave summarized the role of extracellular vesicles in intercellular communications, its diagnostic and prognostic roles. The manuscript is written well, however, authors need to address the following pieces of information. 

RQ1: Authors have described the size of exosomes ranges from 30 nm to 100 nm while several reports suggest the range 30-150 nm (Hoshino et al, 2015 Nat.; Costa-Silva et al 2015 NCB, Patel et al., 2017, British J Cancer, etc).

A1: We corrected the size range of exosomes as suggested, and the related references were added in the text.

RQ2: In the EVs subtype, authors did not describe Apoptotic Bodies and specific markers for the ABs and Microvesicles, if known just like exosomal markers.

A2: Different studies were carried out in order to establish specific phenotypes for the different EV subtypes. To this end, EVs of different size were separated by differential centrifugation procedures. However, since these experiments were performed with different separation approaches and with different EV sources, the international society of extracellular vesicles (ISEV) established that it is not possible to propose specific and universal markers of one or the other type of EVs (PMID: 30637094). For these reasons, in its recent position paper, the ISEV did not propose any molecular marker that could characterize specifically MVs and Apoptotic Bodies. This part has been implemented in the text.

RQ3: Authors also need to describe how MVBs are getting escaped from the lysosomal degradation.

A3: As suggested, the escape of MVBs from the lysosomal degradation has been discussed in the text.

RQ4: How EVs mediate cellular communication? What are the different modes of interaction and internalization of exosomes?

A3: As required, the mechanism mediating EV cellular communication and the different modes of EV interaction and internalization were implemented in the manuscript.

RQ4: How these EVs can be utilized as diagnostic, prognostic and drug carriers?

A4: As suggested, a section discussing how EVs can be used as diagnostic, prognostic and drug carriers has been added.